# Printed Graphene Layer as a Base for Cell Electrostimulation—Preliminary Results

**DOI:** 10.3390/ijms21217865

**Published:** 2020-10-23

**Authors:** Lucja Dybowska-Sarapuk, Weronika Sosnowicz, Jakub Krzeminski, Anna Grzeczkowicz, Ludomira H. Granicka, Andrzej Kotela, Malgorzata Jakubowska

**Affiliations:** 1Faculty of Mechatronics, Warsaw University of Technology, Andrzeja Boboli 8, 02-525 Warsaw, Poland; sosnowicz.weronika@gmail.com (W.S.); krzem@mchtr.pw.edu.pl (J.K.); m.jakubowska@mchtr.pw.edu.pl (M.J.); 2Centre for Advanced Materials and Technologies CEZAMAT, Poleczki 19, 02-822 Warsaw, Poland; 3The Maciej Nalecz Institute of Biocybernetics and Biomedical Engineering, Polish Academy of Sciences, Ksiecia Trojdena 4, 02-109 Warsaw, Poland; agrzeczkowicz@ibib.waw.pl (A.G.); ludomira.granicka@ibib.waw.pl (L.H.G.); 4Faculty of Medicine. Collegium Medicum, Cardinal Stefan Wyszynski University in Warsaw, Dewajtis 5, 01-815 Warsaw, Poland; andrzejkotela@gmail.com

**Keywords:** graphene nanoplatelets, surfactants, cell electrostimulation, tissue engineering

## Abstract

Nerve regeneration through cell electrostimulation will become a key finding in regenerative medicine. The procedure will provide a wide range of applications, especially in body reconstruction, artificial organs or nerve prostheses. Other than in the case of the conventional polystyrene substrates, the application of the current flow in the cell substrate stimulates the cell growth and mobility, supports the synaptogenesis, and increases the average length of neuron nerve fibres. The indirect electrical cell stimulation requires a non-toxic, highly electrically conductive substrate material enabling a precise and effective cell electrostimulation. The process can be successfully performed with the use of the graphene nanoplatelets (GNPs)—the structures of high conductivity and biocompatible with mammalian NE-4C neural stem cells used in the study. One of the complications with the production of inks using GNPs is their agglomeration, which significantly hampers the quality of the produced coatings. Therefore, the selection of the proper amount of the surfactant is paramount to achieve a high-quality substrate. The article presents the results of the research into the material manufacturing used in the cell electrostimulation. The outcomes allow for the establishment of the proper amount of the surfactant to achieve both high conductivity and quality of the coating, which could be used not only in electronics, but also—due to its biocompatibility—fruitfully applied to the cell electrostimulation.

## 1. Introduction

Currently, the possibility to differentiate stem cells from other cells, especially nerve cells, is attracting great attention within academia [1,2,3]. Nerve and glial cells form the nerve tissue; compared to other mammalian body tissues, the nerve tissue has poor regenerative properties. Neurons are made of neural stem cells (NSCs) in the process of neurogenesis [1]; the process takes place in the brain’s neurogenic areas. Therefore, obtaining stem cells for research and cell therapy is limited. The connective tissue—Wharton’s jelly or umbilical cord blood—appears a safe and easily accessible source of adult stem cells which can be used to differentiate into neural cells. Due to the appropriate use of mesenchymal stem cells in vitro, cells with a neural phenotype are obtainable [1]. Owning to the appropriately selected cell stimulation and its parameters in the in vitro culture, it is conceivable not only to initiate the process of cell differentiation into neural cells, but also to control their viability and proliferation [4,5,6,7,8,9]. The tissue engineering to stimulate the growth and differentiation of cells uses cell scaffolds and electrostimulation, using both with direct and alternating electric currents [10,11,12]. As the substrates created with the graphene materials are rough and stiff, the sufficient adhesion of cells to the scaffold is ensured [5,6,13,14].

Printed electronics is one of the fastest growing field of the electronics industry. The development of printed electronics has led to the improvement of a variety of pattern and layer application techniques, making it possible to efficiently coat both stiff and flexible substrates [15,16]. Spray coating is one of the printing techniques which allows for a non-contact and cost-effective way to apply layers and patterns to a wide variety of substrates, especially substrates with a complex geometric structure. The indubitable asset of this technique is also the purity of the process, which eliminates the risk of contamination of the printed material and the layer formed, making it possible to apply created substrates in medical applications. The use of graphene materials in printed electronics methods can enable the production of highly conductive layers and coatings that are non-toxic to mammalian cells, which can be used, for example, in regenerative medicine for cellular electrostimulation [5]. Numerous scientific publications prove the expediency of the graphene-coated platforms for the proliferation and differentiation of induced pluripotent stem cells (iPSC), human mesenchymal stem cells (hMSC), human neural stem cells (hNSC) and preosteoblasts into osteoblasts [6,14,17,18,19]. One of the publications also indicates better proliferation and stronger polarisation of hMSC in the presence of fluorinated graphene; hence, it is possible to control the differentiation of cells to generate neurons [20].

The cellular electrostimulation development, especially on the graphene substrates, is of paramount importance for the regenerative medicine development. However, there is currently no literature describing research utilising highly conductive graphene materials such as graphene nanoflakes. Hence, the application of graphene to biomedical issues is still under discussion among researchers [4]. To date, reports related to the usage of graphene in tissue engineering have mainly focused on carbon materials, e.g., non-conductive graphene oxide (GO) or the reduced graphene oxide (rGO), the reduction of which often complicates the technological process [13,16]. Consequently, there have appeared great expectations associated with the graphene nanoplatelets (GNP)—materials with a sheet structure, the dimensions greater than 100 nm and a thickness of several nanometres—and their use in biomedicine [21]. GNPs are biocompatible with mammalian cells; simultaneously, they maintain graphene’s unique electrical properties [5,22]. GNPs also have beneficial properties, e.g., good thermal conductivity and mechanical strength. They are used to create heterophasic materials in the form of inks and pastes. Our to-date studies confirmed that the developed inks and graphene layers are non-toxic to eukaryotic cells [5]. In the literature, graphene materials have been described as excellent substrates for cell cultures, enhancing the proliferation or differentiation of cells [6,14].

One of the problems emerging in the production of nanocomposite materials based on graphene nanoplatelets is the agglomeration of particles, which has a direct impact on the quality and usability of the created composites. The phenomenon is particularly noticeable in the case of unstable, two-dimensional nanoparticles such as graphene flakes [21,23]. The agglomerates develop as a consequence of Van der Waals forces and physical interactions between the structure layers; accordingly, solution sedimentation takes place [21,24]. In the production of polymer composites, it is imperative to obtain a suspension with the greatest possible homogeneity and stability [13,16]. Due to the homogeneous dispersion of particles in the solution, homogenous surfaces are obtainable [5]. This can be achieved by using an appropriately selected (for a given application) surfactant [25]. Both the type of solvent (whether polar or nonpolar) and the type of dispersed phase (hydrophilic or lipophilic) affect the process of dispersion. The amphiphilic structure of the surfactant may generate all the desired effects described, jet the excess amount, and above the critical concentration of micellization, it will generate the formation of micelles, which are structures that disturb the formed suspension in an undesirable way. To eliminate the unintended effects of the use of the amphiphilic agent, there is a need to properly dose surfactants and check the viscosity, printability and electrical parameters of the resulting suspension [16]. The presence of a surfactant affects some of the rheological properties of the inks, e.g., the surface tension. However, to some extent, it influences the solution viscosity. The selection of an appropriate surfactant and its amount significantly improve the conductivity of the graphene coatings, hence improving the electrostimulation process. Moreover, the addition of the surfactant has an significant effect on the sedimentation of the solution, which affects the stability of the coating depositing in the spray coating process [5].

In the production of inks for biomedical applications, the selection of the right surfactant plays a very crucial role. The surfactant ensures good dispersion of particles in the solution and positively affects ink rheology, but, at the same time, it has to be non-toxic to cells [5]. The surfactant is responsible for the solution homogeneity, layer electrical conductivity and cell behaviour on the coating. Such an approach to the cell stimulation and combining it with printing electronics and nanotechnology (particularly the use of graphene nanoplatelets) is an innovative solution, novelty in the field and was not published before.

In this study, the authors present the methodology of production of highly conductive, biocompatible graphene coatings for applications in cell electrostimulation. The study investigated the influence of surfactant content on the rheological properties of ink, as well as on the conductivity and microgeometry of the layers. The described ink manufacturing methodology and selection of the appropriate component quantity allows for the obtention of inks with the desired viscosity for the spray coating technique (viscosity in the range of 1.3–2.0 mPas). At the same time, the ink forms highly electrical conductive layers (13.5 Ω/□). The printed graphene layers are also characterised with homogeneity, which significantly affects the quality of cellular electrostimulation. The obtained results enable the selection of the ink with the best properties that allow for its effective use, not only in electronics, but, due to its biocompatibility, also for successful use in electrostimulation of cells.

## 2. Results and Discussion

### 2.1. Rheology of Graphene Inks with Different Surfactant Contents

The graphene inks have to meet the rheological requirements determined by the applied printing technique. It is necessary to assess the inks’ viscosity, which affects the quality of the printed surface. On the basis of the obtained results, the relationship between the shear rate and the viscosity of the tested inks was determined. The inks’ viscosity results are shown in Figure 1. The obtained viscosity values of the graphene inks, measured at the shear rate of 487.5 $\frac{1}{S}$, are presented in Table 1.

The serviceability of a material for use in spray coating technology is defined by the viscosity value at the highest shear rate, which should remain within the 0.7–2.0 mPas range [5,9]. The obtained viscosity values of the graphene inks, measured for the shear rate of 487.5 $\frac{1}{s}$, were in the range of 1.4–2 mPas, while for the base ink, the result was 1.3 mPas. The obtained values of viscosity are generally comparable. The viscosity value for the base ink is the lowest, which results from the lack of dispersion of the graphene flakes in the ink solution. Not only does that result in a poor quality of the coating, but it also contributes to its low conductivity. The further viscosity values—for 2%, 5% and 10% surfactant content—are akin to each other. Then, for the inks with 15% and 20% of surfactant, the viscosity value decreases. This indicates a deterioration of the flake dispersion triggered by the too-large amount of the surfactant. The viscosity values of the inks with the surfactants are within the recommended range, which enables their effective application in spray coating technology.

### 2.2. Production of Conductive Graphene Layers by Spray Coating Techniques

Spray coating allows one to place the graphene layers on variously shaped surfaces. It is also possible to achieve the desired thickness of the layers in a relatively easy manner. Additionally, the contactless application and high cleanliness of the coating are important, particularly for the biomedical applications.

During the application of the coatings, significant differences between several inks were established. The base ink, containing no surfactant, contained the agglomerates of the graphene flakes which clogged the airbrush nozzle; this significantly hindered the entire process and resulted in the production of nonhomogeneous coatings. Moreover, the coatings with such an ink required more layers to achieve satisfactory homogeneity. The printing process of the remaining—surfactant enriched—inks went smoothly regardless of the surfactant content.

Figure 2 and Figure 3 show examples of the Kapton film substrates and polystyrene culture plates: (a) without a layer; (b) with a layer made with the base ink; and (c) ink with a surfactant. Figure 3 shows the flexible Kapton film substrates before and after the graphene coatings made with inks containing 0% and 5% surfactant. It can be noticed that the substrate created with the base ink is of poor quality, and produced agglomerates are clearly visible.

The quality of the obtained layers proves the surfactant’s effectiveness. Analysing the above photos, it is evident how great the impact of the surfactant content in the ink is on the coating process. Not only does the surfactant component enable the correctness of the printing, but also, it unquestionably improves the quality of the produced layers, visible already during the layer application.

### 2.3. Layer Micro- and Macro-Geometry

From the biological perspective, due to their impact on cells and their stimulation, both micro- and macro-geometry are central. Figure 4 and Figure 5 display the enlargements of the created graphene layers to assess the surfaces’ micro- and macro-geometry, depending on the surfactant content.

The analysis of the micro-geometric images of the coatings proves that their porosity decreases with the increase in the surfactant content. The base ink coating has numerous defects, perceptible without high magnification. The addition of a little amount of surfactant significantly improves the coating homogeneity and reduces porosity. However, increasing the surfactant content from 5% to 20% provided no significant improvement in the coating quality. All the produced layers are equally homogeneous and lack pores. The increasing surfactant content insignificantly improves the quality of the microgeometry. The exception is the coating produced with the 15% surfactant ink, which has poor quality areas. Yet, in that case, it is more likely an effect of the distortion of the substrate rather than the poor quality of the produced ink itself.

Taking into account the cytocompatibility of the AKM-0531 surfactant (D2), confirmed in our previous work [25], the inks made with it can successfully be used in the cell cultures. The examination of the coatings’ micro- and macro-geometry showed no significant variances for coatings made with inks containing more than 5% of surfactant. The most significant differences in the coatings’ quality can be observed when comparing the SEM photographs of the coatings made with the base ink and the ink containing with the 5% surfactant. The addition of even a small amount of surfactant has a profound impact on the coatings’ uniformity and quality and, thus, on their electrical conductivity.

### 2.4. Conductivity of Graphene Layers

To enable the application of the produced graphene coatings in the cellular electrostimulation, it is crucial to obtain the best current conductivity. The conductivity of the layers was measured depending on the amount of the surfactant used. The results of sheet resistivity measurements and their representation in figures are presented in Table 2 and Figure 6.

The graph’s analysis indicates significant differences among the sheet resistivity values for various surfactant contents. When the surfactant content increases from 0% to 10%, the resistivity decreases successively; then, for the 15% and 20% content, their increase is visible. The lowest resistivity values were established for the 10% surfactant ink. Interestingly, the values are even five times lower than those obtained for the base ink (without any surfactant). Obtaining such low resistance proves the paramount influence that creating a homogeneous ink with a well-dispersed functional phase has on the layers’ conductivity. The obtained viscosity results straightforwardly influence the coatings’ conductivity (described in Section 2.1). The addition of more than 10% caused no further decrease in the resistance values; contrary, the values actually increased. The reason may be the bigger distance between the graphene nanoflakes, as a result of which, the conductivity of the layers decreased.

Choosing the spray coating method for applying the coatings allowed for the desired surface quality and conductivity to be achieved, while maintaining high purity of the process. The spray coating method allows for the obtention of graphene rough substrates, which can positively affect the behaviour of cells and their adhesion.

### 2.5. Cell Electrostimulation—Preliminary Results

For the applications of the graphene coatings in tissue engineering, cytocompatibility of the inks towards the cells is essential. There is a potential risk of the ink’s toxicity resulting from the use of certain surfactants in the heterophasic inks [5]. The low content surfactant inks show no toxicity to the cells. Still, increasing the amount up to 20% could, potentially, have a negative effect on the cell behaviour. Accordingly, it is recommended to use less surfactant, which reduces the danger of cytotoxicity; simultaneously, it ensures good electrical conductivity. Therefore, the initial electrostimulation was performed on the layers made of the 5% surfactant content ink.

The preliminary electrostimulation process, carried out with the use of the produced substrates, resulted in the formation of new connections and cellular extensions. The contact of the cells with the graphene substrate did not cause their death; however, further research is necessary to determine more precise substrate–cell interactions. The SEM observations have made it possible to assess the ability of the proper adherence and elongation of the cells cultured on graphene layers for a seven-day culture. Figure 7 presents the results for both the current-stimulated and unstimulated control cells.

The SEM pictures show that the electrostimulation greatly increased the number of correctly flattened cells. The cells from the control culture properly adhere to the substrate, flatten out and sprout. Still, when the current is applied, the well-developed network, created by the cell appendages, can be observed. Moreover, the electrostimulated cells released appendages, which, in the case of neurons, create ovules of dendrites and axons.

Based on microscopic observation, it can be concluded that there is an observable effect of electrostimulation on the behaviour of the neuronal stem cells embedded in the graphene layer. Electrostimulated cells not only multiply decidedly faster, but also form numerous cylindrical cytoplasmic appendages, which are extensions of the nerve cell body.

## 3. Materials and Methods

The composition of the inks was selected on the basis of the results obtained in our previous research [25]. The graphene nanoplatelets manufactured by XG Science (XG Science, Inc.; Lansing, MI, USA) (GNP M25) with average particle dimensions of 20–25 μm and a thickness of 19 sheets of graphene were used as the material of the functional phase [26]. The material was also used and characterised in our previous studies [27,28]. The percentage content of graphene flakes was 0.5 wt.%. The matrix is an essential component of graphene ink. In our study, a solution of polymeric polymethyl methacrylate (PMMA) (Sigma-Aldrich Sp.zo.o.; Poznan, Poland) in butyl carbinol acetate (OKB) (Sigma-Aldrich Sp.zo.o.; Poznan, Poland) at a concentration of 8% was used. Due to the long dissolution time of PMMA in OKB (min. 24 h), the matrix solution was conducted as the first step. The process of homogenisation of the carrier was carried out on a magnetic stirrer with a heated plate model RTC produced by IKA (IKA Poland Sp. zo.o.; Warsaw, Poland) at a temperature of 50 °C. Each of the produced inks contained the same matrix material in content of 27.5 wt.% and the same solvent—acetone, in content of approximately 72 wt.%. The percentage contents of each component in the made solution are presented in Table 3 below.

The AKM-0531 surfactant MALIALIM series from the NOF company (NOF America Corporation; White Plains, NY, USA) was used in the studies. The MALIALIM surfactant is made of functional polymers with a comb structure. According to the datasheet, the molecular weight of the surfactant is in the range of $10^{4}–10^{5}$ u [29]. MALIALIM surfactants have ionicity groups on the main chains responsible for the absorption of the polymer to the powder surface and its wetting. The polyoxyalkylene groups on side chains cause the repulsive effect of the functional phase particles, improving their dispersion in suspension [30].

In regard to the percentage content of the graphene flakes, inks containing 2, 5, 10, 15 and 20 wt.% surfactant were made. Concentrations of the surfactant with respect to the whole ink solution are presented in Table 4 below. Previous studies have shown that the layers produced with the developed inks have not caused toxic effects against cells [25]; thus, it seems certain that the layers used in the current work are cytocompatible.

The layers were printed on two different substrates: a 25-µm-thick flexible Kapton film (DuPont, Inc.; Wilmington, DE, USA) and the polystyrene cell culture plates from Nest Scientific Biotechnology (Wuxi Nest Scientific Co., Ltd.; Wuxi, Jiangsu, China). The graphene inks were made according to the technology presented in Figure 8. The ink base was a combination of the surfactant, the graphene flakes and acetone. The obtained ink base was sonicated in an InterSonic IS-10 (Intersonic Sp.zo.o.; Olsztyn, Poland), 700 W ultrasonic cleaner for 90 min carried out at 35 kHz. After sonication, a polymer was added to the ink base. Finally, the whole solution was sonicated again for 15 min.

The layers were produced by spray coating. The Harder & Steenbeck Infinity CR Plus Solo Airbrush (Harder and Steenbeck, GmbH & Co. KG; Norderstedt, Germany) was used for application. The airbrush was connected to a compressed gas cylinder with 0.3 MPa pressure. The airbrush nozzle’s diameter was 400 µm. The distance between the nozzle and the substrate was approximately 20 cm. After the application, the coatings were dried in a Memmert UF55 laboratory dryer with natural air circulation (Memmert GmbH & Co. KG; Schwabach, Germany. The drying of the Kapton foil substrates was carried out at 120 °C and lasted 60 min. For polystyrene culture plates, the drying temperature was 70 °C, and the drying time was 240 min.

For the viscosity measurements of the manufactured inks, a Wells-Brookfield model DV2T cone-plate viscometer (AMETEK Brookfield, Inc.; Middleboro, MA, USA) was used with a type CP-40 cone. The coatings’ resistance measurements were performed using the UNI-T digital laboratory multimeter, model UT804 (Uni-Trend Technology, Co., Ltd.; Shanghai, China). The micro- and macro-geometry of the obtained layers, their quality, and homogeneity were assessed with the Keyence VHX-900F digital microscope (Keyence International (Belgium), NA/SA; Mechelen, Belgium), and the scanning electron microscope used was the Phenom Pharos Desktop Scanning Electron Microscope (Thermo Fisher Scientific; Waltham, MA, USA). The pictures were taken at 20×, 100× and 200× magnification and an acceleration voltage of 10 kV.

The NE-4C neuroectodermal cell line was maintained in Eagle’s minimum essential medium (EMEM) (ATCC) supplemented with 10% heat-inactivated foetal bovine serum (FBS) (ATCC), 2 mM L-Glutamine (Sigma-Aldrich Sp.zo.o.; Poznan, Poland) and 10^−6^ M all-trans retinoic acid (RA) (Sigma Aldrich) in the culture flasks pre-coated with 15 µg/mL poly-L-lysine (Sigma Aldrich Sp.zo.o.; Poznan, Poland). NE-4C cells from the 8th to 11th passage were used in the study. The cells were seeded on the 6-well plates at a density of 2 × 10^3^ /cm^2^ and maintained in standard cell culture conditions (37 °C, 5% CO_2_ and 95% humidity). The cell stimulation was performed using an array of comb electrodes. The selected parameters were as follows: voltage amplitude U = 10 V; forcing frequency f = 1 kHz; sinusoidal; alternating current. The stem neural cells were cultured for 7 days; the stimulation started 24 h after seeding the culture [5]. For the scanning electron microscopy observation, after the seven-day culturing, the cells were fixed in 4% glutaraldehyde. Having been dehydrated, the cell surface of the samples was covered with a thin layer of gold with a sputtering system. The specimens were observed with an Merlin Zeiss scanning electron microscope (SEM) (Carl Zeiss Microscopy GmBh, Jena, Germany) with an acceleration voltage of 5 kV.

## 4. Conclusions

The results confirmed the effectiveness of the applied surfactant. The results for the inks with various contents of the AKM-0531 surfactant from the MALIALIM series are in the range of 11–50 Ω/□. The low resistance values of the produced layers enable their effective use not only in cell electrostimulation, but also for the production of sensors requiring high electrical conductivity.

The results prove that the created graphene inks meet both the technical and biological requirements. The use of a surfactant increases the dispersion of the GNPs particles in the inks and improves their homogeneity, which is crucial for obtaining high-quality substrates. The high quality of the substrates directly influences the layers’ electrical properties, which are essential in the electrostimulation process. Obtaining the conductive, homogenous graphene coatings contributes to the tissue engineering development.

However, to entirely exclude the possibility of long-term damage of the cellular structures, further studies concerning the influence of the size and shape of graphene structures on their potential cytotoxicity are necessary [9]. Owning to the development of the research using the mesenchymal stem cells and the neural cells produced with them, in the future, it appears probable to effectively and safely treat injuries and neurodegenerative diseases [14]. In the long term, the use of cellular electrostimulation may provide a solution to the currently incurable neurological conditions, e.g., reconstruction of the spinal cord and nerve connections possibilities.

## Figures and Tables

**Figure 1 ijms-21-07865-f001:**
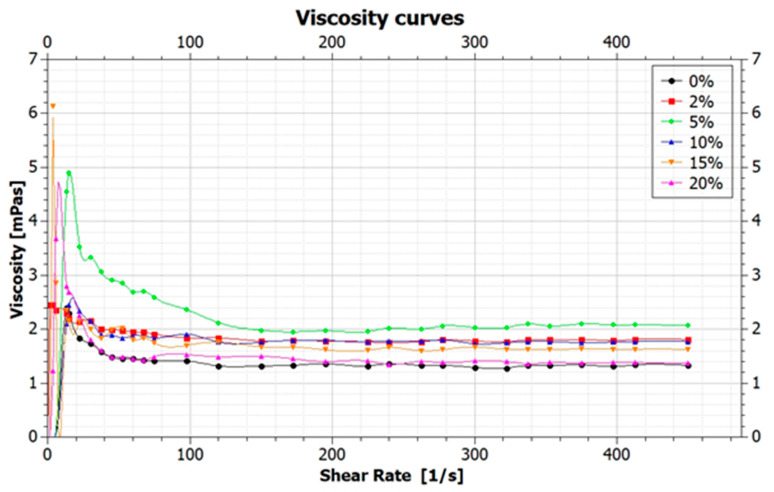
Graphene viscosity curves for graphene inks with 0, 2, 5, 10, 15, 20% surfactant content (in regard to GNP content).

**Figure 2 ijms-21-07865-f002:**
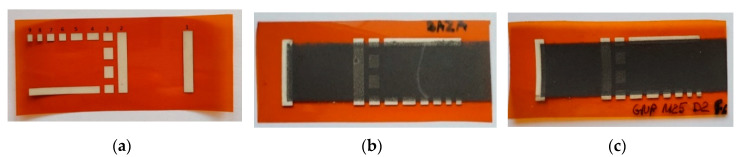
Kapton film substrates (**a**) without a layer; (**b**) with the layer made with the base ink with a visible agglomerate; and (**c**) with the homogeneous layer made with an ink with 5% surfactant content.

**Figure 3 ijms-21-07865-f003:**
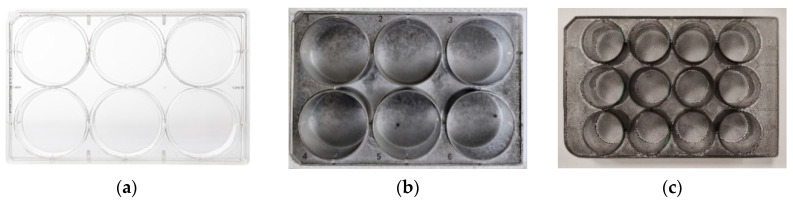
Culture plate (**a**) without coating; (**b**) covered with a base ink coating (without surfactant); and (**c**) covered with an ink coating containing 5% surfactant.

**Figure 4 ijms-21-07865-f004:**
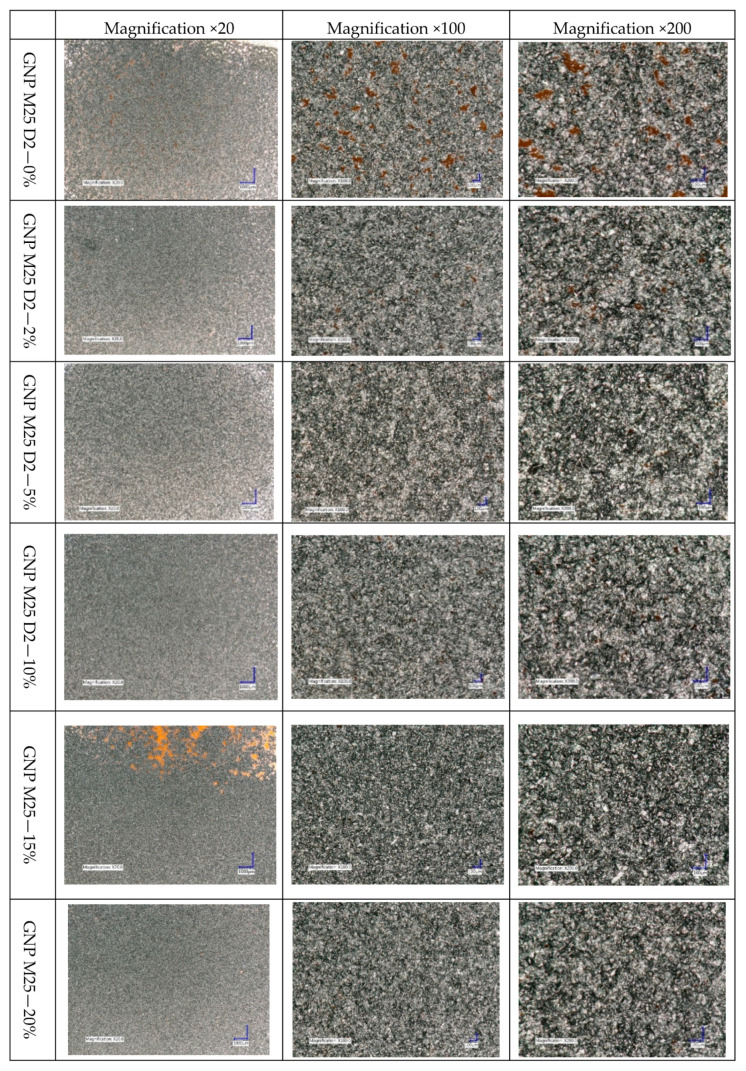
Digital microscope photographs showing the quality of coatings made of the inks with various surfactant content.

**Figure 5 ijms-21-07865-f005:**
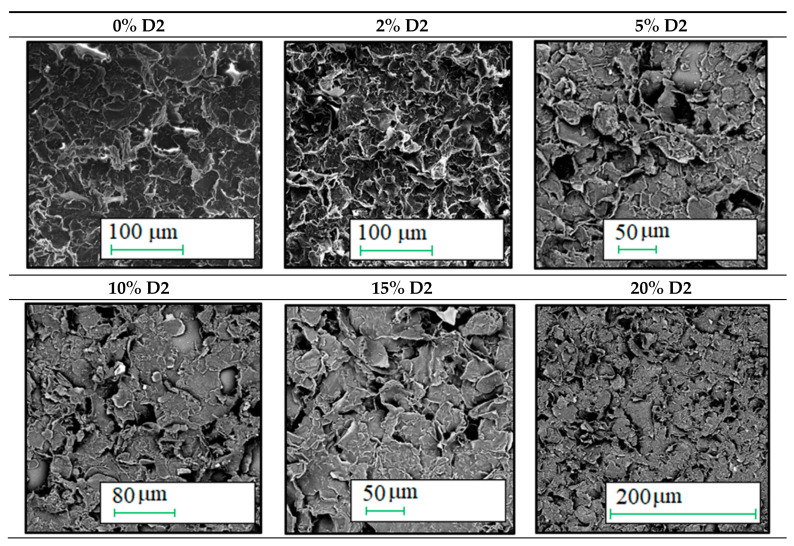
SEM photographs showing the quality of coatings made of the inks with various surfactant content.

**Figure 6 ijms-21-07865-f006:**
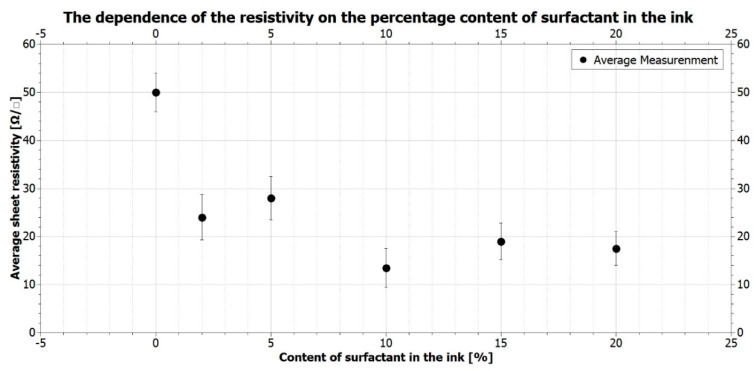
The dependence of the resistivity on the percentage content of the surfactant in the ink.

**Figure 7 ijms-21-07865-f007:**
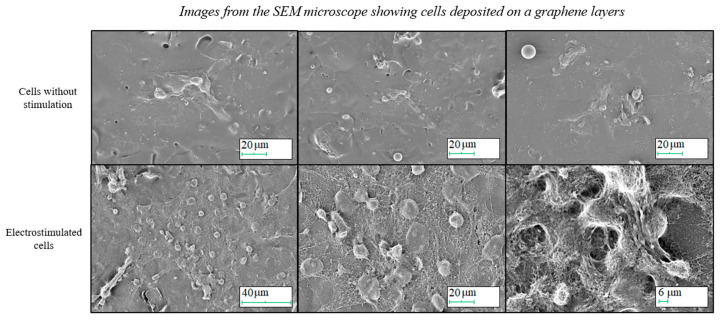
The SEM microscope images presenting the preliminary electrostimulation results.

**Figure 8 ijms-21-07865-f008:**
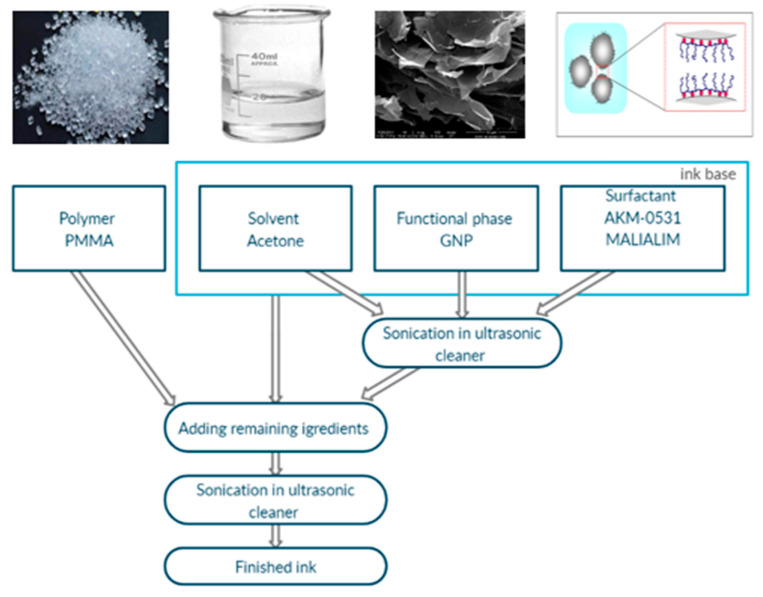
The technology of the graphene inks with graphene nanoflake production [31,32,33,34].

**Table 1 ijms-21-07865-t001:** The obtained values of the viscosity of graphene inks with 0, 2, 5, 10, 15, 20% surfactant content (in regard to GNP content), measured at the highest value of shear rate of 487.5$\;\frac{1}{s}.$

Ink (Graphene Nanoplatelets/Surfactant Content)	Viscosity Value at Shear Rate 487.5 $\frac{1}{s}$
GNP D2 0%	1.3
GNP D2 2%	1.8
GNP D2 5%	2.0
GNP D2 10%	1.8
GNP D2 15%	1.6
GNP D2 20%	1.4

**Table 2 ijms-21-07865-t002:** The results of resistivity of the graphene layer measurements.

Average Sheet Resistivity [Ω/□]					
0% D2	2% D2	5% D2	10% D2	15% D2	20% D2
50.0	24.0	28.0	13.5	19.0	17.5

**Table 3 ijms-21-07865-t003:** Percentage content of ink components.

		Ink Components		
Percentage Content of Surfactant in Regard to GNP Content [%]	Surfactant (%)	Graphene Nanoplatelets (%)	Matrix (%)	Solvent (%)
0	0	0.5	27.5	72.000
2	0.01	0.5	27.5	71.990
5	0.025	0.5	27.5	71.975
10	0.05	0.5	27.5	71.950
15	0.075	0.5	27.5	71.925
20	0.1	0.5	27.5	71.900

**Table 4 ijms-21-07865-t004:** Weight content of the surfactant in the ink composition for a 50 g sample.

Ink					
	GNP D2 2%	GNP D2 5%	GNP D2 10%	GNP D2 15%	GNP D2 20%
Content of the surfactant (g)	0.005	0.0125	0.025	0.0375	0.05

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
