# Peer review of "Printed Graphene Layer as a Base for Cell Electrostimulation—Preliminary Results"

_ijms, 2020, doi:10.3390/ijms21217865_

Round 1
Reviewer 1 Report
In this article by Dybowska-Sarapuk et al., the authors described how they used printed graphene layers as a base for neural cell growth facilitated by electrostimulation. Clearly, this work fits an interesting and hot branch of research regarding the application of nanomaterials for cell growth. However, the evaluated manuscript is not at the appropriate level to enable the publication of these findings in the International Journal of Molecular Sciences. I invite the authors to consider the following suggestions to improve it. Upon incorporation of the corrections, the paper may be re-evaluated.
1) Language - the work contains many errors, which should be corrected. It is suggested to proofread the file to eliminate them. What is more, the narration is unclear e.g. " The surfactant works by disturbing the structure of the solvent (by the lyophobic part of the molecule), breaking its bonds and creating its own structures near the hydrophobic part." (Lines 78-80). That is true if the solvent is for instance water and the dispersed solid is hydrophobic. It is recommended to follow the reasoning and correct the whole contribution to be more exact.
2) Formatting - the manuscript is very disorganized. Firstly, "." should be the decimal point. Secondly, the formatting of references is very unconventional (e.g. [1]–[3], Line 36). Thirdly, the SEM micrographs do not contain professional scale bar markers. The images from the digital microscope have some illegible text. Please note that it was not noticed that the text in Fig. 7 was underlined in red. The formatting of the plots is not consistent, etc.
3) Introduction - the novelty factor is not clearly defined. Please specify exactly what is new in this work as compared with the state of the art. It would also be beneficial to have a paragraph detailing what was done in this study at the end of the introduction section. It is a common practice, which facilitates navigation in the paper for readers.
4) "Until now, literature reports related to the usage of graphene in tissue engineering mainly concern on carbon materials, such as non-conductive graphene oxide (GO), or reduced, often under difficult conditions, graphene oxide (rGO)" (Lines 58-61). It is an exaggeration since even such a simple reductant as vitamin C can be used to reduce one into the other.
5) Materials and methods - this section lacks scientific rigor. The essential details regarding the used materials and techniques are missing from the file. Firstly, graphene should be characterized by at least Raman spectroscopy, SEM, etc. What is the number of layers? This can have a major impact on the generated findings, but it was neglected. Secondly, the processing conditions are not precisely reported. For example (there are more errors, please locate them and make the necessary amendments):
- "The percentage content of graphene flakes was 0.5 wt. %. Each of the produced inks contained the same matrix material, which was a polymer solution of polymethyl methacrylate (PMMA) in butyl carbinol acetate (OKB) at the concentration of 8% and the same solvent - acetone. The content of the ingredients was sequentially 27.5 wt. % and 72 wt. %." (Lines 224-227) - from this description it is not possible to reason out the concentration of all the components.
- "The AKM-0531 surfactant from the MALIALIM series was used in the production of all inks." (Line 228) - what exactly is MALIALIM? What is the structure? What is the molecular weight?
- "The inks containing 2, 5, 10, 15 and 20 wt. % surfactant in regard to the percentage content of graphene flakes were made." (Lines 233-234) Please report what was the concentration of surfactant with respect to the whole mixture. It does not make sense to specify it with regards to graphene.
- Sonication - it is more important to specify the power/amplitude rather than the frequency
- The model of the airbrush and the nozzle size are missing
- "The drying process of the Kapton foil substrates was carried out at the temperature of 120°C and lasted one hour. For polystyrene culture plates the drying temperature was 70°C and the drying time was 4 hours." (Lines 248-250) Only the drying temperature was reported. How about the conditions?
- what was the SEM acceleration voltage?
- etc.
6) Results. Three critical flaws of this work are as follows:
- there is no convincing proof that graphene is any better than e.g. graphite. It is suggested to reproduce this study by taking graphite as a reference or e.g. other types of carbon or nanocarbon (e.g. carbon nanotubes). Why is graphene the best material to do the job?
- lack of statistics - individual points are plotted, with no error bars, and then these results are interpreted. Without error analysis, it is impossible to say if these results are statistically significant. This, in turn, puts in doubt the conclusions based on them.
- "The layers were printed on two different substrates: a 25 μm thick flexible Kapton film and polystyrene cell culture plates from Nest Scientific Biotechnology" (Lines 237-238). Acetone was used as a medium, but unfortunately, polystyrene has poor compatibility with acetone, so most probably these plates were damaged in the coating process. It cannot be excluded that this changed the roughness and released some of the components. Please make a comment on this issue.
Overall, in light of what was said, it seems as if the manuscript was composed without paying the necessary attention. Had the authors reviewed the documentation before the submission, they would have noticed that one of the paragraphs is composed of instruction for authors "Materials and Methods should be described with sufficient details to allow others to replicate and build on published results. Please note that publication of your manuscript implicates that you must make all materials, data, computer code, and protocols associated with the publication available to readers. Please disclose at the submission stage any restrictions on the availability of materials or information. New methods and protocols should be described in detail while well-established methods can be briefly described and appropriately cited." (Lines 251-256). I strongly suggest ensuring that the revised version matches the level of precision presented in other works published by the International Journal of Molecular Sciences
Reviewer 2 Report
The manuscript describes graphene layer as a base for cell electrostimulation. As the preliminary results, the manuscript can be accepted after minor revision.
Authors have given a brief description of the ink property and the printing results in Results and discussion part. But I think in Introduction part should give some introduction about printed electronics and emphasize the importance of the printing technology. Some review papers can be cited: https://onlinelibrary.wiley.com/doi/full/10.1002/admt.201800546 and https://pubs.rsc.org/no/content/articlelanding/2018/cs/c8cs00084k/unauth#!divAbstract
Round 2
Reviewer 1 Report
Revisions completed satisfactorily. The article can be published as is.